

# Enhancement of *E. coli* acyl-CoA synthetase FadD activity on medium chain fatty acids

Tyler J. Ford[1] and Jeffrey C. Way[2]

[1] Department of Systems Biology, Harvard Medical School, Boston, MA, USA
[2] Wyss Institute for Biologically Inspired Engineering, Harvard Medical School, Boston, MA, USA

## ABSTRACT

FadD catalyses the first step in *E. coli* beta-oxidation, the activation of free fatty acids into acyl-CoA thioesters. This activation makes fatty acids competent for catabolism and reduction into derivatives like alcohols and alkanes. Alcohols and alkanes derived from medium chain fatty acids (MCFAs, 6–12 carbons) are potential biofuels; however, FadD has low activity on MCFAs. Herein, we generate mutations in *fadD* that enhance its acyl-CoA synthetase activity on MCFAs. Homology modeling reveals that these mutations cluster on a face of FadD from which the co-product, AMP, is expected to exit. Using FadD homology models, we design additional FadD mutations that enhance *E. coli* growth rate on octanoate and provide evidence for a model wherein FadD activity on octanoate can be enhanced by aiding product exit. These studies provide FadD mutants useful for producing MCFA derivatives and a rationale to alter the substrate specificity of adenylating enzymes.

## INTRODUCTION

Medium chain fatty acids (MCFAs, 6–12 carbons) are important precursors to fuel-like compounds and industrial chemicals (*Handke, Lynch & Gill, 2011*; *Knothe, 2009*). *E. coli* have been engineered to produce MCFAs using a variety of techniques (*Akhtar et al., 2015*; *Choi & Lee, 2013*; *Dehesh et al., 1996*; *Dellomonaco et al., 2011*; *Torella et al., 2013*; *Voelker & Davies, 1994*), but their conversion into fuel-like compounds such as alcohols and alkanes requires activation of the MCFA carboxylic acid head group into a stronger electrophile. Biologically, this can be achieved by converting the carboxyl group into an acyl-CoA thioester. The acyl-CoA synthetase FadD catalyses this conversion in *E. coli* aerobic beta-oxidation and has been used to activate long chain fatty acids (LCFAs, 13+ carbons) for their later reduction into fuel-like compounds (Fig. 1) (*Black et al., 1992*; *Doan et al., 2009*; *Steen et al., 2010*; *Zhang, Carothers & Keasling, 2012*). However, FadD has low activity on fatty acids less than 10 carbons long resulting in slow *E. coli* growth rates on these fatty acids even in the presence of mutations de-repressing *fadD* and other genes involved in beta-oxidation (*Campbell, Morgan-Kiss & Cronan, 2003*; *Iram & Cronan, 2006*; *Kameda & Nunn, 1981*; *Overath, Pauli & Schairer, 1969*; *Salanitro & Wegener, 1971*).

Corresponding author
Tyler J. Ford, tyjoford@gmail.com

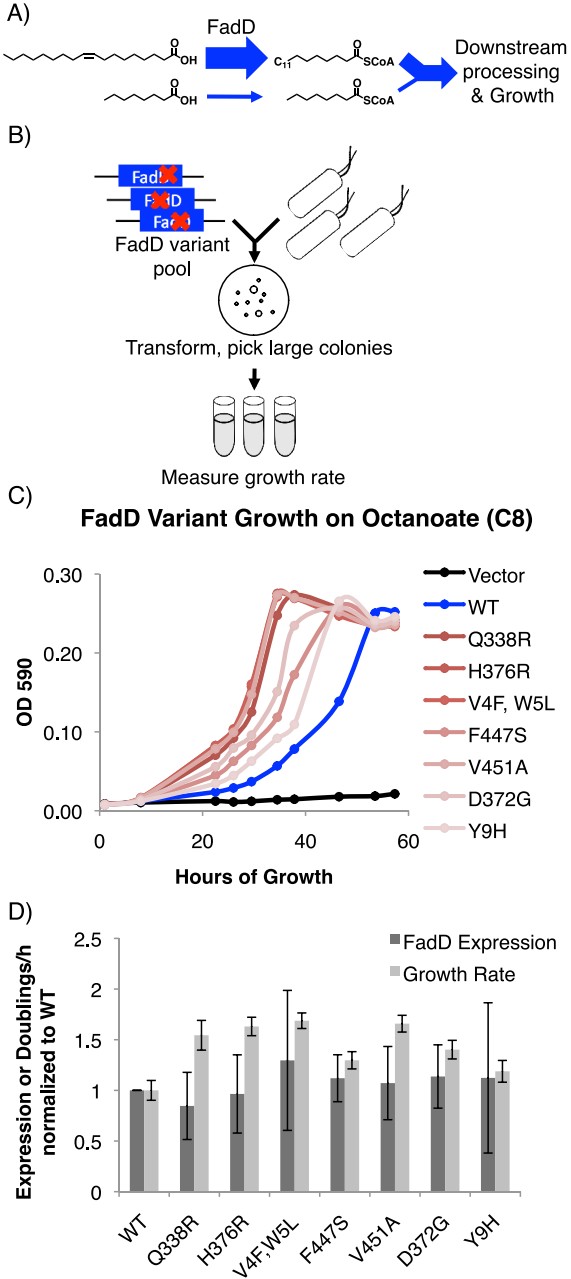

**Figure 1** **FadD mutants generated by error prone PCR increase *E. coli* Δ*fadR* growth rate on octanoate without increasing FadD expression.** (A) FadD catalyzes the first step in *E. coli* growth on fatty acids but has low activity on fatty acids shorter than 10 carbons. (B) Error prone PCR and FadD screening scheme (Materials and Methods). (C) Growth of *E. coli* Δ*fadR* expressing the indicated *C*-terminally $His_6$-tagged FadD mutants generated by error prone PCR from vector pETDuet-1 on octanoate. (D) Relative increase in FadD expression (dark gray) and growth rate (light gray) conferred by $His_6$-tagged FadD mutants on octanoate compared to wild-type FadD. $n = 5$ for FadD expression and 6 for growth rate; error bars indicate standard deviation. All increases in growth rate have $p < 0.05$ by two sided students *T*-test while all changes in expression have $p > 0.3$. FadD expression was measured using anti-his western blot samples normalized to total protein content by $A_{280}$ (Materials and Methods).

*Salmonella enterica*, which has a FadD very similar to that of *E. coli*, grows more quickly than *E. coli* on octanoate, but this is due to changes in *fadD* regulation and the activity of downstream beta-oxidation enzymes and not to changes in FadD enzymatic activity (*Iram & Cronan, 2006*).

The mechanisms behind FadD substrate specificity are not well understood. This protein belongs to a class of adenylate-forming enzymes for which numerous crystal structures have been solved, including an LCFA-specific, FadD homolog from *Thermus thermophilus* (*Conti, Franks & Brick, 1996*; *Conti et al., 1997*; *Gulick, 2009*; *Gulick et al., 2003*; *Hisanaga et al., 2004*; *Hu et al., 2010*) and an MCFA-specific homolog from *Homo sapiens* with butyryl-CoA and AMP in the active site (*Kochan et al., 2009*). Analyzing the structure of *Thermus thermophilus* acyl-CoA synthase co-crystalized with myristoyl-AMP, *Hisanaga et al. (2004)* hypothesized that, because the myristoyl-AMP rests in a tunnel well-suited to accommodate its long, hydrophobic tail, it is the length of this tunnel that determines substrate specificity. A similar mechanism has been used to explain the medium chain specificity of the human homolog (*Kochan et al., 2009*), but it has not been shown whether decreasing or increasing the size of this tunnel experimentally can alter substrate specificity. *Black et al. (1997)* constructed mutations in a conserved fatty acyl-CoA synthetase (FACS) motif in FadD. These had subtle effects on FadD selectivity, but only one showed an absolute increase in activity on decanoate (*Black et al., 1997*). The FACS motif is adjacent to a region of FadD involved in fatty acid binding, but no further mutagenesis studies of this region have led to increased FadD activity on MCFAs shorter than 10 carbons (*Black et al., 2000*).

Herein, we specifically enhanced FadD activity on MCFAs shorter that 10 carbons using a strategy incorporating *fadD* mutagenesis by error prone PCR and a growth-based screen for acyl-CoA synthetase activity. We hypothesized that FadD mutants that enhance *E. coli* growth rate on octanoate would have increased activity on MCFAs because FadD catalyzes the first step in fatty acid catabolism. We generated FadD mutants that confer increased growth rate on the MCFAs hexanoate (6-carbons), octanoate (8-carbons), and moderately on decanoate (10-carbons), but not palmitate (16-carbons) or oleate (18-carbons). *In vitro* assays of partially purified wild-type FadD and mutant variants showed that they possess increased activity on octanoate and decanoate, but not oleate. Homology modeling revealed that the isolated mutations cluster around a proposed AMP exit channel from the FadD active site (*Hisanaga et al., 2004*; *Kochan et al., 2009*), and mutations designed to widen this exit channel confer increased growth rate on octanoate. These FadD mutants show that it is possible to alter the substrate specificity of acyl-CoA synthetases without necessarily improving substrate binding and may provide a rationale to engineer other adenylate-forming enzymes important for processes ranging from lignin processing (*Hu et al., 2010*) to antibiotic production (*Conti et al., 1997*).

## MATERIALS AND METHODS

### Error Prone PCR and fadD mutant screening

Error prone PCR mixtures contained 90 μl Go-Taq Green 2X Master Mix (Promega, Madison, Wisconsin, USA) mixed with ∼150 ng of TJF032 (pETDuet-1 containing wild type *fadD*) template, 0.5 μM each of forward and reverse primers, 40 μM $MnCl_2$ and $H_2O$ to 180 μl. The resulting PCR products were digested with NcoI and HindIII and ligated into pETDuet-1 (Novagen, Madison, Wisconsin, USA). Ligation products were transformed into BW25113 Δ *fadR* with a separate TJF032 control and plated on octanoate minimal medium (M9 + 1 g/L octanoate, 0.2% Nonidet$^{TM}$ P 40 Substitute [Sigma] 15% agar), containing 50 μg/mL ampicillin (Amp). Transformants grew for 3 days at 37 °C. Colonies larger than those on the TJF032 plate were restreaked on octanoate minimal plates along with TJF032 transformant controls, allowed to grow for 3 further days, and restreaked a second time. Transformant colonies larger than TJF032 transformant colonies after this third streak were picked into 5 mL LB/Amp, grown overnight, and miniprepped. Miniprepped constructs were sequenced using primers TF0017 and TF0018, re-transformed into JW1176-1, and transformants plated on LB/Amp. 6 colonies from each transformation as well as 6 colonies from a TJF032 control transformation were then picked into 1 mL LB/Amp each in a 96 well deep well plate (Nunc, Rochester, New York, USA). Cultures were grown overnight (∼18 h) and then diluted 1:50 into 1 mL M9 octanoate containing 50 μg/mL Amp. The $OD_{590}$ of each culture was monitored throughout growth in octanoate minimal medium using a Victor 3v Multilabel Plate Reader (Perkin Elemer, Waltham, Massachusetts, USA). Doublings and doublings/h were determined by diving all $OD_{590}$ values by the $OD_{590}$ recorded 1 h after dilution, calculating the log in base 2 of this value, and plotting this against hours of growth. The slope of the linear portion of this curve ($R^2 > 0.9$) was recorded as the doublings per hour. Growth rates on hexanoate (0.90 g/L), decanoate (0.80 g/L), palmitate (0.74 g/L), and oleate (0.73 g/L) were determined similarly. Palmitate and oleate minimal media had 0.4% NP40, 0.2% ethanol, and 1% Triton X-100 to solubilize the fatty acids.

### Western blotting

Strain JW1176-1 expressing the appropriate C-terminally His$_6$-tagged FadD variant was grown on M9 octanoate minimal medium as described above. 1 mL samples were taken from early exponential phase cultures and boiled in 3% SDS. Total protein concentration was normalized by $A_{280}$ and samples were western blotted with an HRP conjugated antibody to the His$_6$ tag (ab1187, abcam) diluted 1:10,000 in TBS-tween with 1% BSA. Relative band intensities of the FadD variants in their linear range (as determined by serial dilutions) were quantified in Image J (*Schneider, Rasband & Eliceiri, 2012*).

### Site directed mutagenesis

Site directed mutants were constructed using the QuikChange II Site-Directed Mutagenesis Kit (Agilent, Santa Clara, California, USA) and plasmid TJF032 as template per the manufacturers' instructions. Successfully generated constructs were sequenced and

transformed into JW1176-1 and growth rates in octanoate minimal medium measured as indicated for the error prone PCR mutants above.

## Partial purification of C-terminally His$_6$-tagged FadD

For all purifications, the appropriate C-terminally His$_6$-tagged FadD variants were purified from BL21*(DE3) $\Delta$fadD. Fresh transformation mixtures were diluted directly into 5 mL LB containing 50 µg/mL Amp. Cultures were grown overnight at 37 °C with shaking at 250 rpm and back diluted 1:500 into 250 mL LB/Amp in 1 L Erlenmeyer flasks at 21 °C with shaking at 250 RPM. After 13 h of growth at 21 °C (OD$_{600}$ ~0.2), cultures were induced with 0.1 mM IPTG and incubated for 9 further hours. Cells were then harvested by centrifuging at 4,000 × $g$ for 10 min at 4 °C in a J6-M1 centrifuge (Beckman Coulter, Brea, California, USA) (all buffers and incubations for the remainder of the procedure were at 4 °C). The cell supernatant was then poured off and the pellet resuspended in 10 mL lysis buffer (50 mM NaH$_2$PO$_4$, 300 mM NaCl, 10 mM imidazole, pH 8) with 1 mg/mL lysozyme (Sigma, St. Louis, Missouri, USA), 0.125 mg/mL DNase I (Sigma, St. Louis, Missouri, USA), 1 µg/mL pepstatin, and 1 protease inhibitor cocktail tablet for general use (Sigma, St. Louis, Missouri, USA). The resuspended pellet was then sonicated in a 550 Sonic Dismembrator (Fisher Scientific, Waltham, Massachusetts, USA). Samples were centrifuged at 14,000 × $g$ for 30 min in an Avanti J-301 centrifuge (Beckman Coulter, Brea, California, USA). After centrifugation, the cell lysate (supernatant) was transferred to a new tube, the pellet discarded, and 5 µl of the lysate added to 5 µl 2X Tris-Glycine SDS sample buffer (Life Technologies, Carlsbad, California, USA) and stored at room temperature for later analysis by SDS-PAGE. FadD-His in the lysate was then bound to 200 µl NiNTA beads (Qiagen, Hilden, Germany). Beads were washed twice with 4 mL of wash buffer (50 mM NaH$_2$PO$_4$, 300 mM NaCl, 20 mM imidazole, pH 8.0) and eluted twice in 1 mL of elution buffer (50 mM NaH$_2$PO$_4$, 300 mM NaCl, 500 mM imidazole). 5 µl of flow through, wash, and elution samples were taken as above to monitor the purification. Eluate was then transferred to an Amicon® Ultra-15 Centrifugal Filter Ultracel® with 30 kDa molecular weight cut off (Millipore, Billerica, Massachusetts, USA). Samples were centrifuged at 4,000 rpm for 15 min in a bench top Centrifuge 5,810 R (Eppendorf, Hamburg, Germany). The flow through was discarded, 12 mL buffer C (20 mM Tris-HCl, 150 mM NaCl, pH 8.0) added to the column, and the process repeated twice. Samples were centrifuged similarly a final time, resuspended in 2 mL buffer C, TCEP added to a final concentration of 5 mM, and stored at 4 °C overnight for kinetic analysis the next day or glycerol added to a final concentration of 20% and the samples flash frozen in liquid nitrogen and stored at −80 °C. All samples in 1X Tris-Glycine sample buffer were then visualized by SDS-PAGE and Coomassie stained to ensure proper purification (Fig. S1).

## Kinetic analysis of partially purified Acyl CoA synthetase (FadD)
### AMP production assay

Kinetic assays coupling the FadD catalyzed production of acyl-CoAs and AMP from oleate and octanoate to the oxidation of NADH were monitored spectrophotometrically via

measuring absorbance at 340 nm in a Synergy NEO HTS Multi Mode microplate reader (BioTek) (*Kameda & Nunn, 1981*). Reactions were carried out at 30 °C in 100 μl total of freshly prepared 20 mM Tris-HCl pH 7.5, 2.5 mM ATP, 8 mM $MgCl_2$, 2 mM EDTA, 0.1% Triton X-100, 0.5 mM CoA, 0.2 mM NADH, 0.3 mM phosphoenolpyruvate (PEP), 48 U Myokinase from Chicken Muscle (Sigma, St. Louis, Missouri, USA), 96 U Pyruvate Kinase From Rabbit Muscle (Sigma, St. Louis, Missouri, USA), 48 U of Lactic Dehydrogenase (Sigma, St. Louis, Missouri, USA), 0.2 μg Ni-NTA purified FadD, and the appropriate amount of fatty acid from 1,000 X stock solutions in ethanol (Oleate concentrations: 2.66–170 μM, Octanoate concentrations: 15.0–964 μM). Reactions were initiated with the addition of CoA and absorbance at 340 nM was measured every 30 s for 10 min. To ensure that the reactions were limited by FadD and not by the coupled enzymes, prior to measuring the activities of other purified enzymes, kinetics of wild-type FadD with oleate as substrate were determined with 0.4, 0.2, and 0.1 μg FadD. If the oleate $V_{max}$ increased proportionally with the amount of enzyme, it was assumed the coupling enzymes were not limiting.

### Acyl-CoA production assay

Acyl-CoA production assays directly measured the production of 14-C fatty acyl-CoAs based on fatty acyl-CoA partitioning into an aqueous phase vs. organic phase after the CoA synthetase reaction (*Kameda & Nunn, 1981*). Assay mixtures contained 1.6 μg of Ni-NTA purified FadD (or the appropriate mutant), 0.05 M Tris-HCl pH 8.0, 0.01 M $MgCl_2$, 0.01% Triton X-100, 10 mM ATP, and 0.3 mM DTT in 1 mL. Radiolabled fatty acids were included at a final concentration of 0–1.5 mM for octanoate, 100 μM for oleate, and 50 μM for decanoate. Reactions were initiated via the addition of 200 μM CoA and 0.25 mL periodically transferred to separate tubes containing 1.25 mL stopping buffer (40:10:1 isopropanol:n-heptane:1M $H_2SO_4$) to terminate the reaction. The terminated mixtures were then extracted 3 times with 1 mL n-heptane and radioactivity in 200 μl of the remaining aqueous phase measured by liquid scintillation counting in an LS6500 Multi-purpose Scintillation Counter (Beckman Coulter, Brea, California, USA). Counts determined in this way were plotted over time and standards containing known amounts of fatty acid were used to determine the counts/nmol fatty acid. Slopes of the counts *v* time plots were then converted to nmol fatty acid/time giving the acyl-CoA production rate.

### $K_m$ and $V_{max}$ determination

Once enzymatic rates were determined for each FadD preparation, rates were plotted against the concentration of fatty acid substrate used in each reaction and curves were fit to the Michaelis Menten equation using the nlinfit fuction in MATLAB Release 2010b (The Mathworks, Inc., Natick, Massachusetts, United States).

$$V = V_{max}[x]/(K_m + [x]).$$

Where $x$ is the concentration of fatty acid and $V$ is the rate of acyl-CoA production determined as indicated above. Only curves with $R^2$ values >0.9 were accepted (Fig. S2).

To normalize kinetic assay results for protein purity, prior to running either assay, wild-type FadD and its variants were visualized by SDS-PAGE and Coomassie staining. The full-length FadD bands were then quantified in ImageJ (*Schneider, Rasband & Eliceiri, 2012*). Wild-type FadD band intensities were used to adjust all mutant protein concentrations used for rate determinations by the relative intensity of each full-length mutant band to the intensity of the full-length wild-type FadD band.

## TSS competent cell preparation and transformation

All transformations were performed according to the TSS competent cell protocol described previously (*Chung, Niemela & Miller, 1989*).

## Homology modeling

FadD homology models were generated using The SWISS-MODEL Homology modeling server (*Arnold et al., 2006*; *Benkert, Biasini & Schwede, 2011*; *Biasini et al., 2014*) and the *Thermus thermophilus* structure as the template, the I-TASSER server (*Roy, Kucukural & Zhang, 2010*; *Yang et al., 2015*; *Zhang, 2008*), and (iii) SAM-T08 (*Karchin, Cline & Karplus, 2004*; *Karchin et al., 2003*; *Karplus, 2009*; *Karplus & Hu, 2001*; *Karplus et al., 2001*; *Karplus et al., 2003*; *Karplus et al., 2005*; *Shackelford & Karplus, 2007*). Models were visualized in Mac Pymol (The PyMOL Molecular Graphics System, Version 1.7rc1 Schrödinger, LLC.) and Swiss-PdbViewer (*Guex & Peitsch, 1997*).

## RESULTS

### Mutations generated in the FadD coding sequence increase *E. coli* growth rate on MCFAs but not LCFAs

*fadD* mutants generated by error prone PCR confer increased *E. coli* growth rate on octanoate. We generated mutations in the *fadD* coding sequence using error prone PCR and screened mutants for their ability to increase *E. coli* Δ*fadR* (a strain with constitutively active β-oxidation) growth rate on octanoate (Fig. 1, Materials and Methods). Plasmids from strains with increased growth rate over controls were isolated and sequenced. In total, seven FadD single mutants conferred increased growth rate on octanoate Fig. 1B.

The FadD mutants generated by error-prone PCR do not increase FadD expression. To ensure that the FadD mutants do not increase growth rate by simply enhancing FadD expression, wild-type FadD and the FadD mutants were C-terminally His$_6$-tagged, growth was measured (Fig. 1B), and SDS-PAGE samples were prepared at early exponential phase (~26 h of growth in octanoate minimal medium). Samples were normalized for total protein by A$_{280}$ and western blotted with an anti-his antibody (Materials and Methods). While increases in growth rate were very consistent ($p < 0.05$ in all cases), there was no significant difference in FadD expression between the wild-type and mutant variants (Fig. 2C).

FadD mutants increase growth rate on the MCFAs hexanoate, octanoate, and moderately on decanoate, but do not increase growth rate on the LCFAs palmitate and oleate. To determine whether the effects of these mutations, selected on octanoate, were specific to octanoate or were more broadly effective on fatty acids of different chain lengths, we measured their effects on growth rate in hexanoate (C6), decanoate (C10), palmitate

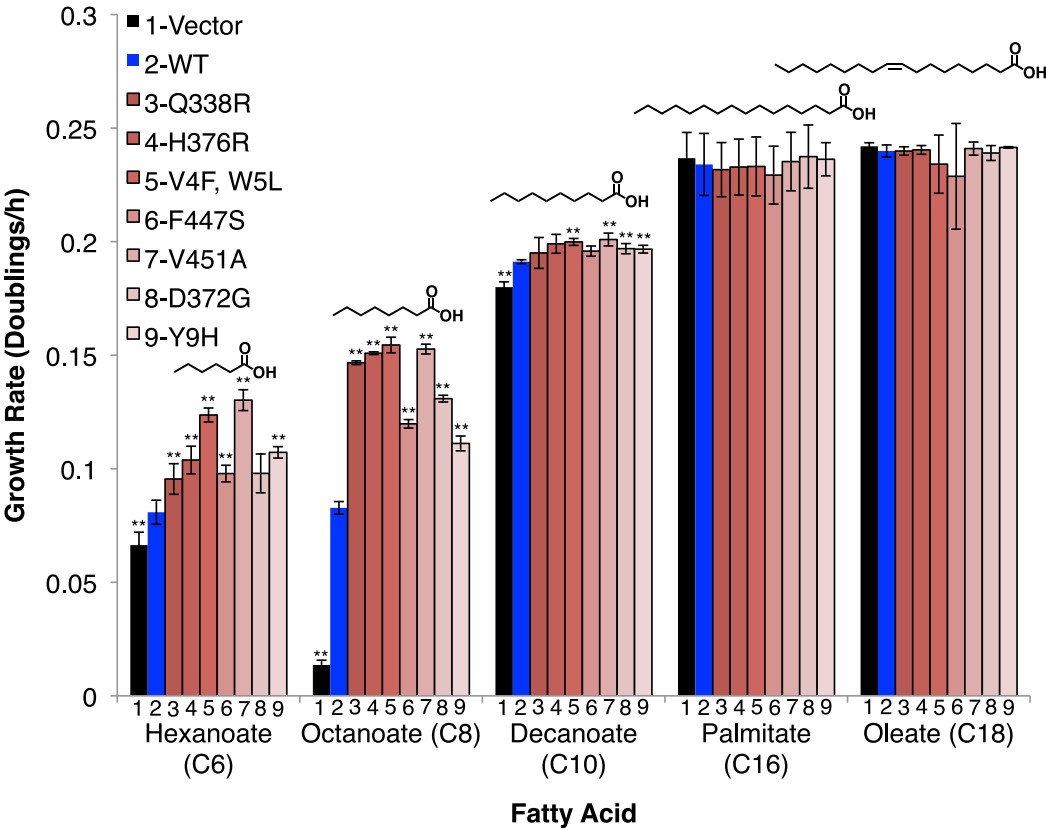

**Figure 2** **FadD mutants enhance the growth rate of *E.coli* Δ*fadR* on the MCFAs hexanoate, octanoate, and decanoate, but not on palmitate and oleate.** *E.coli* Δ*fadR* transformed with empty pETDuet-1 (black) *C*-terminally $His_6$-tagged wild-type *fadD* (blue) or the indicated *C*-terminally $His_6$-tagged *fadD* mutants (Red) were grown on minimal medium containing the indicated fatty acid as the sole carbon source. Growth rates were measured by linear regression of the normalized $\log_2$ (OD590) during exponential phase. $n = 3$, errors bars indicate standard deviation, and ** indicates $p < 0.05$ compared to wild-type by two sided students $T$-test.

(C16), and oleate (C18) minimal medium. The mutants had strong effects on hexanoate and octanoate medium, but only marginally increased growth rate on decanoate and failed to alter growth rate on palmitate, or oleate (Fig. 2).

## FadD mutant proteins have increased activity on octanoate and decanoate, but not oleate

*In-vitro* assays measuring AMP production by the FadD mutants showed that they have increased activity on MCFAs but not LCFAs. The assay coupled AMP production in the acyl-CoA synthetase reaction to the oxidation of NADH which was monitored spectrophotometrically (Materials and Methods) (*Kameda & Nunn, 1981*). The $V_{max}$ values of the mutants were higher than those of wild-type FadD on octanoate, but were generally lower on oleate (with the exception of mutant H376R). There were no significant

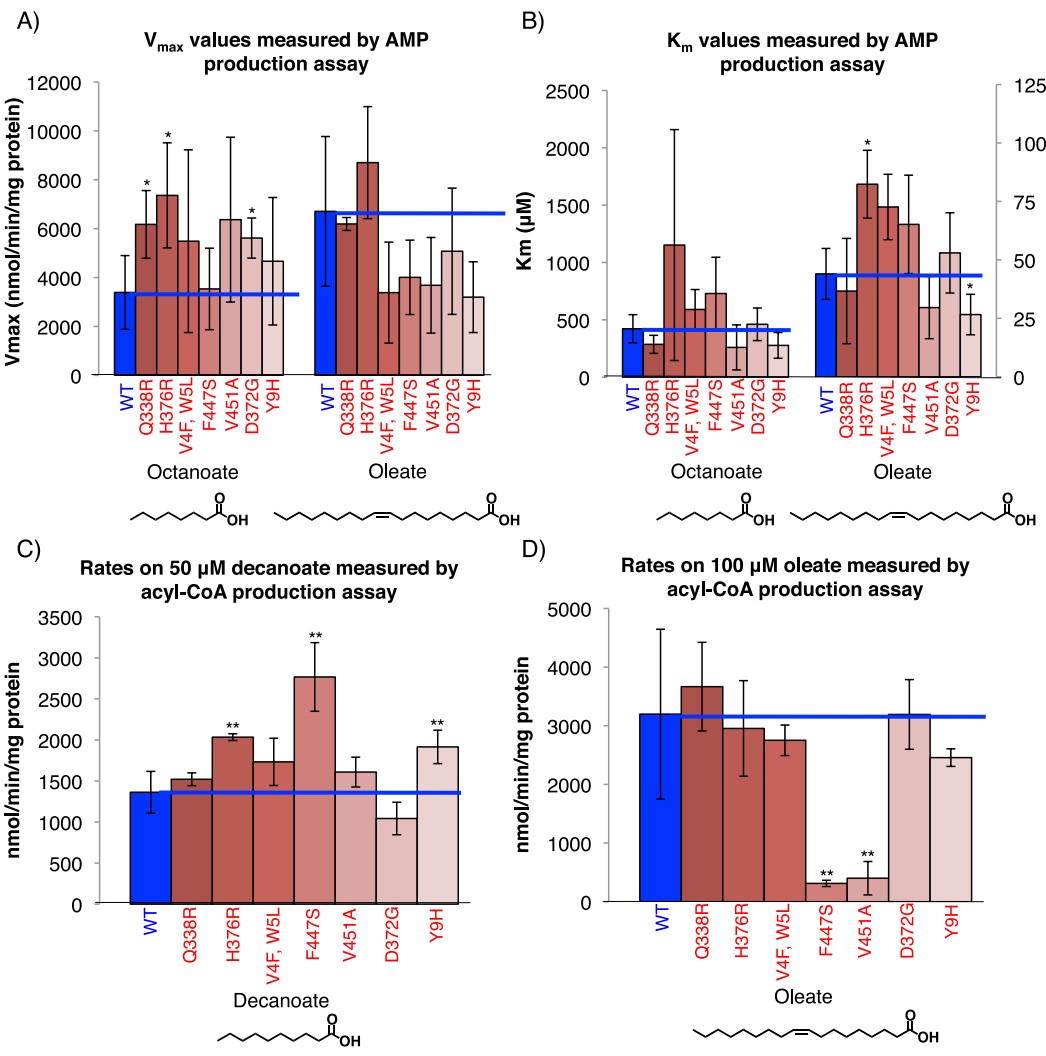

**Figure 3 FadD mutants have increased activity on MCFAs but unaltered affinity for MCFAs.** (A) $His_6$-tagged wild-type FadD and the indicated mutants were partially purified via Ni-NTA purification (Materials and Methods) and steady state activity on the indicated fatty acid measured spectrophotometrically using the AMP production assay (Materials and Methods) (*Kameda & Nunn, 1981*). $V_{max}$ (A) and $K_m$ (B) values for the indicated substrates. $n = 3$–4 independent purifications, error bars indicate standard deviation, * indicates $p < 0.1$ compared to wild-type by two sided students $T$-test. (C) and (D) Steady state rate of acyl-CoA production using 1.6 µg of Ni-NTA purified, C-terminally $His_6$-tagged wild-type FadD and the indicated mutants with decanoate (C) and oleate (D) as substrates at concentrations roughly 10 times the literature reported $K_m$ values in the acyl-CoA production assay (Materials and Methods) (*Kameda & Nunn, 1981*). $n = 3$ independent measurements from one or two purifications, error bars indicate standard deviation, and ** indicates $p < 0.05$ by two sided student's $T$-test compared to wild-type FadD.

changes in the $K_m$ toward octanoate for each of the mutants, although the mutant H376R showed an increased $K_m$ toward oleate while Y9H had a decreased $K_m$ toward oleate (Figs. 3A and 3B). These results indicate that, while the FadD mutations increase the rate of the acyl-CoA synthetase reaction, they do not generally enhance FadD affinity for octanoate.

**Table 1 Catalytic efficiencies of FadD mutants.** Catalytic efficiency ($K_{cat}/K_m$) of wild-type FadD and the indicated mutants as measured by AMP production assay.

| FadD variant | Octanoate $K_{cat}/K_m$[a] | Oleate $K_{cat}/K_m$[a] |
|---|---|---|
| WT | $0.10 \pm 0.07$ | $1.70 \pm 0.94$ |
| Q338R | $0.25 \pm 0.13$ | $1.98 \pm 1.07$ |
| H376R | $0.10 \pm 0.06$ | $1.16 \pm 0.54$ |
| V4F, W5L | $0.10 \pm 0.05$ | $0.46 \pm 0.20$ |
| F447S | $0.05 \pm 0.02$ | $0.74 \pm 0.50$ |
| V451A | $0.35 \pm 0.28$ | $1.46 \pm 1.01$ |
| D372G | $0.14 \pm 0.05$ | $1.09 \pm 0.65$ |
| Y9H | $0.20 \pm 0.07$ | $1.38 \pm 0.80$ |

**Notes.**

[a] ($M^{-1} * s^{-1} * 10^5$) Values are indicated $\pm$ standard deviation.

Calculated catalytic efficiencies ($K_{cat}/K_m$) (Table 1) show that all mutants except Q338R were less efficient than wild-type when oleate was used as a substrate, but four of the mutants (Q338R, V451A, D372G and Y9H) were more efficient than wild-type when using octanoate. The remainder of the mutants had lower or equivalent catalytic efficiency on octanoate indicating that decreases in affinity toward octanoate (higher $K_m$) outweighed or matched increases in overall activity (higher $K_{cat}$).

A second *in-vitro* assay directly measuring acyl-CoA production showed that the mutants have increased activity on decanoate and octanoate but not oleate. Rates determined using decanoate and oleate as substrates at concentrations roughly 10 times their published $K_m$ values (*Kameda & Nunn, 1981*) in the acyl-CoA production assay (Material and Methods) showed that, while most of the mutants have increased activity on decanoate, none have significant increases in activity on oleate and two have decreased activity (Figs. 3C and 3D). This is consistent with the data from the AMP production assays.

FadD mutant proteins had higher maximal activity on octanoic acid in acyl-CoA production assays, consistent with the AMP production assays (data not shown). However, the rates determined in the acyl CoA production assays using octanoate as substrate had high background and poor fits to the Michaelis–Menten curve. Presumably the high background activity was due to the higher solubility of octanoate as compared to oleate or decanoate, which both gave lower background and more consistent measurements.

## Site directed FadD mutants designed to open a proposed AMP exit channel increase *E.coli* $\Delta fadR$ growth rate on octanoate

FadD homology models generated using the SWISS-MODEL Homology modeling server (*Arnold et al., 2006*; *Benkert, Biasini & Schwede, 2011*; *Biasini et al., 2014*) and the *Thermus thermophilus* structure as the template, the I-TASSER server (*Roy, Kucukural & Zhang, 2010*; *Yang et al., 2015*; *Zhang, 2008*), and SAM-T08 (*Karchin, Cline & Karplus, 2004*; *Karchin et al., 2003*; *Karplus, 2009*; *Karplus & Hu, 2001*; *Karplus et al., 2001*; *Karplus et al., 2003*; *Karplus et al., 2005*; *Shackelford & Karplus, 2007*), show that several of the

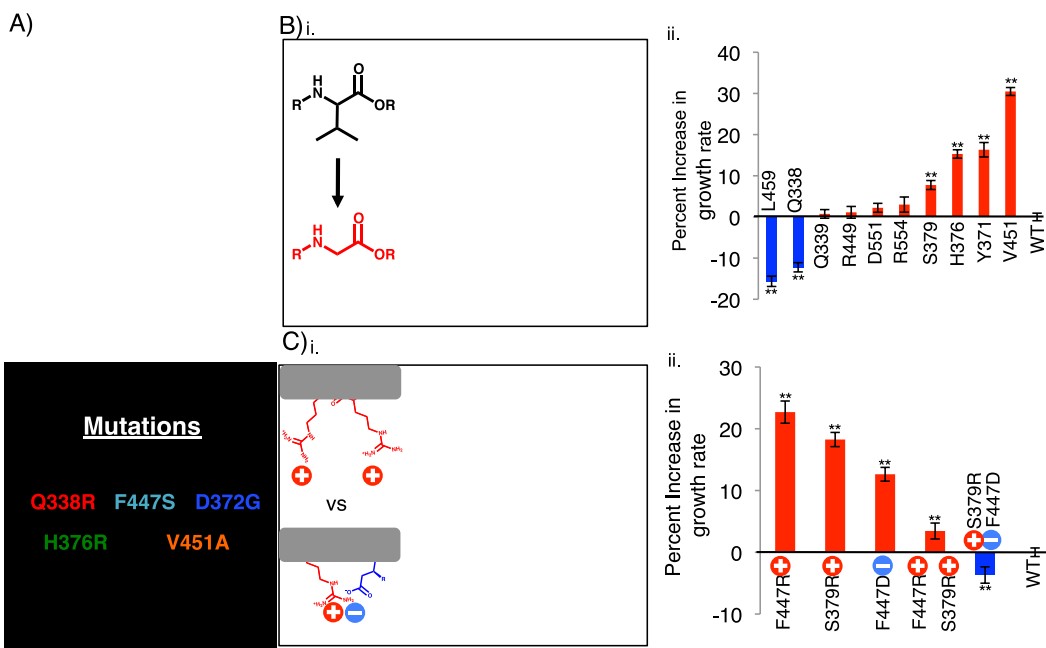

**Figure 4 Rationally designed, site directed FadD mutants increase *E. coli* Δ*fadR* growth rate on octanoate when compared to wild-type FadD.** (A) FadD homology model generated using The SWISS-MODEL Homology modeling server (*Arnold et al., 2006*; *Benkert, Biasini & Schwede, 2011*; *Biasini et al., 2014*) and the *Thermus thermophilus* structure as the template. The model was visualized in PyMOL with large *N*-terminal domain in gray, smaller *C*-terminal domain in white, and myristoyl-AMP (overlayed from the *Thermus thermophillus* structure) in yellow (myristoyl group) and magenta (AMP) (*Hisanaga et al., 2004*). Residues whose mutation results in increased growth rate on octanoate are color-coded according to the identity of the mutation (text below model, Y9H and V4F W5L are excluded from the model). (B) (i) Surface representation of the FadD homology model with residues mutated to glycine in (ii) shown in blue (mutations that decrease growth rate compared to wild-type) and red (mutations that increase growth rate compared to wild-type). (ii) Percent increase in exponential growth rate compared to wild-type FadD caused by mutating the residues on the *X*-axis to glycine. (C) (i) Cartoon representation of FadD homology model with residues mutated in (ii) in red. (ii) Percent increase in exponential growth rate compared to wild-type FadD caused by the FadD mutations depicted on the *X*-axis. $n = 13$–$18$, error bars indicate standard error in all cases, ** indicates growth rate significantly different from wild-type with $p < 0.05$ by two-sided students *T*-test.

FadD mutations cluster around a possible ATP/AMP entrance/exit channel (Fig. 4A and Fig. S3). All models have features similar to those of known adenylating enzymes as well as the acyl-CoA synthetase from *Thermus thermophilus* (*Conti, Franks & Brick, 1996*; *Conti et al., 1997*; *Gulick, 2009*; *Gulick et al., 2003*; *Hisanaga et al., 2004*; *Hu et al., 2010*). These include a small, globular C-terminal domain (white), a large, globular N-terminal domain (grey), and an active site (annotated by the alignment in *Hisanaga et al. (2004)*) situated between the two domains. Comparing these homology models to the structure of the *Thermus thermophilus* acyl-CoA synthetase shows that several of our FadD mutations cluster on a face of the protein from which ATP and AMP are proposed to enter and exit the active site (*Hisanaga et al., 2004*). *Hisanaga et al. (2004)* inferred that ATP binding precedes and enhances fatty acid binding, so enhancement of ATP binding would likely decrease the $K_m$ for the fatty acid. Given that our mutants fail to decrease $K_m$, but do increase $V_{max}$

toward octanoate, we hypothesize that they could facilitate AMP exit from the active site by opening this face of the protein.

FadD mutations designed to facilitate AMP exit from the FadD active site increased growth rate on octanoate (Fig. 4). To test the hypothesis that opening the FadD AMP exit channel could facilitate product exit and increase FadD activity on MCFAs, we removed amino acid side chains surrounding the channel by mutating their associated residues to glycine, and measured the resultant mutants' growth on octanoate. Eight out of ten of these mutations increased the average growth rate of *E.coli* $\Delta fadR$ (JW1176-1) compared to wild-type. Two of these ten mutations decreased growth rate (Fig. 4B).

Further mutations designed to electrostatically repel structurally adjacent amino acids (S379, F447) and thereby destabilize the closed confirmation of FadD and aid AMP exit enhanced growth rate on octanoate. Mutations designed to electrostatically attract these same amino acids decreased growth rate on octanoate. The mutations were made separately and in combination and their effects on the growth rate of *E.coli* $\Delta fadR$ (JW1176-1) on octanoate minimal medium were measured. Specifically, residues S379 and F447, which are adjacent to each other in the FadD homology model (Fig. 4Ci), were each mutated to arginine and aspartate singly and in combination. Each individual mutation and the double mutant designed to repel these residues and destabilize the closed confirmation of FadD (S379R, F447R) enhanced growth rate on octanoate. In contrast, the double mutant designed to form a salt bridge between these residues and stabilize the closed confirmation of FadD (S379R, F447D) decreased growth rate on octanoate.

## DISCUSSION

This work shows that *E. coli* FadD activity limits the conversion of medium chain fatty acids (MCFAs; 6–12 carbons) to acyl-CoA thioesters and provides a set of FadD mutants that will be useful in expediting this conversion. We identified mutations in the *E. coli* K12 *fadD* gene by constructing a library of altered genes via PCR mutagenesis, transformation, and screening for enhanced growth on octanoic acid (Fig. 1). The mutant genes significantly increased the host growth rate on hexanoate and octanoate, somewhat on decanoate, and not at all on palmitate or oleate (Fig. 2). Kinetic assays indicated that the FadD mutant proteins have an increased $V_{max}$ toward octanoate, without significant effects on $K_m$. These results suggest that these mutations increase activity without enhancing substrate binding. Given that our FadD mutants were screened on 6.9 mM octanoate minimal medium, a concentration far in excess of the wild-type FadD $K_m$ for octanoate determined here (422 µM), it is perhaps unsurprising that mutations conferring higher affinity for octanoate were not discovered.

Although measurements of the FadD mutants' enhanced acyl-CoA synthetase activities *in vivo* and *in vitro* differed somewhat, differences can likely be explained by two factors: (1) differences in lipid composition *in vivo* and *in vitro* and (2) activities of downstream beta-oxidation enzymes *in vivo*. FadD activity is enhanced by both the presence of membrane lipids and detergents (*Mangroo & Gerber, 1993*). Although we added Triton X-100 to our *in vitro* assay mixtures (materials and methods), it is likely that interactions

between Triton X-100 and FadD do not perfectly mimic interactions between FadD and the *E. coli* membranes resulting in differences in the activity observed *in vivo* and *in vitro*. With particular reference to the data in Figs. 2 and 3 showing little enhancement in growth rate on decanoate, but statistically significant enhancement of activity toward decanoate *in vitro* (mutant H376R for instance), discrepancies such as these are likely due to the limitations of downstream beta-oxidation enzymes *in vivo*. Although enhanced FadD activity generates more acyl-CoAs, little increase in growth rate is observed because downstream beta-oxidation enzymes, which have poor activity on medium chain acyl-CoAs (*Iram & Cronan, 2006*), become limiting.

The mechanism of FadD is complex and involves multiple substrate-binding and product exit steps through different channels in the protein. *Hisanaga et al. (2004)* solved a structure of a FadD homologue from *Thermus thermophilus*, with and without an AMP-fatty acid intermediate. Based on these structures and prior biochemistry, they proposed that, as the FadD protein is a non-integral membrane-associated protein, the fatty acid enters from the membrane through a narrow channel in the back of the protein, while ATP enters through a distinct, large channel. ATP and the fatty acid bind first and form the AMP-fatty acid intermediate, releasing pyrophosphate. At this point, a flexible C-terminal domain, clamps onto the AMP-fatty acid intermediate to prevent its escape and position it for nucleophilic attack by CoA, which then binds and attacks the phosphoester bond, generating AMP and fatty acyl-CoA. *Kochan et al. (2009)* determined the structures of a human medium-chain acyl-CoA synthetase with ATP and butyryl-CoA/AMP in the active site. The pantotheine group of CoA enters by a third channel in the protein, distinct from the ATP and fatty acid entry sites.

When mapped onto a homology model of FadD, our mutations are nowhere near the binding sites of either the fatty acid or CoA, but some border on the ATP/AMP channel and amino acids that may directly or indirectly affect the interaction of the flexible C-terminal domain with the rest of the protein; these include Val 451, which may make direct contact and Asp 372, His 376, Gln 338, and Phe 447, which may indirectly affect the structure of the AMP exit channel or the interaction of this region of the protein with the C-terminal domain. Additionally, none of our mutations fall in a region (residues 422–430) involved in fatty acid binding as shown by affinity labeling experiments (*Black et al., 2000*). We hypothesize that when CoA bonds to a long-chain fatty acid, the AMP is sterically pushed from the active site by this product. When CoA bonds to an MCFA on the other hand, it may move within the active site so that this steric push is less pronounced. The effect of these mutations might be to ease the transition to an open state and enhance AMP exit, which would result in the observed increase in $V_{max}$.

There is a second acyl-CoA synthetase in *E. coli*, FadK, which has higher relative activity on MCFAs than LCFAs, but has lower absolute activity than FadD (*Campbell, Morgan-Kiss & Cronan, 2003*; *Morgan-Kiss & Cronan, 2004*). We specifically chose to focus on FadD because of its higher absolute activity. Endogenous *fadK* may have contributed somewhat to the growth and acyl-CoA synthetase activity observed here, but any contribution is expected to be minimal as FadK is only expressed under anaerobic conditions

(*Campbell, Morgan-Kiss & Cronan, 2003*; *Morgan-Kiss & Cronan, 2004*) and all experiments here were performed under aerobic conditions.

This work adds to our growing knowledge on structural determinants of FadD substrate specificity. One of the mutations discovered here (V451A) falls in the previously characterized FACS motif (*Black et al., 1997*). Our results confirm that this mutation increases FadD activity on decanoate (as shown previously) and further show that this mutation enhances activity on octanoate and hexanoate. The remainder of our mutations falls outside this motif. This agrees well with our kinetic data that the mutant proteins have unchanged $K_m$ values and thus likely do not enhance MCFA binding. Future work screening FadD mutants on lower MCFA concentrations could produce FadD mutants with decreased $K_m$ toward MCFAs.

Some of our mutants have unaltered activity on oleate (Q338R and H376R) while others have decreased activity toward oleate (F447S and V451A). These two types of mutants could prove useful for different reasons. For example, future work requiring a mixture of acyl-CoA lengths would benefit from the mutants with high activity on both MCFAs and LCFAs, while work specifically producing only medium chain products would benefit from the mutants with decreased activity on LCFAs and increased activity on MCFAs.

More broadly, there are many adenylate-forming enzymes like FadD that have similar structures and functions but modify different substrates (*Conti, Franks & Brick, 1996*; *Conti et al., 1997*; *Gulick, 2009*; *Gulick et al., 2003*; *Hu et al., 2010*). The method of increasing activity on a similar but smaller substrate by aiding product exit may be applicable to other adenylate-forming enzymes with similar structures. This approach could open possibilities for engineering the degradation and modification of a variety of substrates important for applications ranging from lignin processing (*Hu et al., 2010*) to antibiotic production (*Conti et al., 1997*).

## ACKNOWLEDGEMENTS

We thank Joseph Torella, Stephanie Hays, Paul Black, and Jessica Polka for critical reading of the manuscript. We thank Paul Black, Stephen Hinshaw, and Gabriel Birrane for helpful advice on protein purification and enzymatic assays.

### Funding

This work was conducted with support from the Advanced Research Projects Agency-Energy 'Electrofuels' Collaborative Agreement DE-AR0000079, the National Science Foundation Graduate Research fellowship (to T.J.F.), and the Ruth L. Kirschtein National research Service Award program of Harvard Catalyst, The Harvard Clinical and Translational Science Center Award UL1 RR 025758 and financial contributions from Harvard University and its affiliated academic health care centers (to T.J.F.). The content is solely the responsibility of the authors and does not necessarily represent the official views of Harvard Catalyst, Harvard University and its affiliated academic health care centers, the National Center for Research Resources or the National Institutes of Health. This

material is based upon work supported by the National Science Foundation. Any opinions, findings, and conclusions or recommendations expressed in this material are those of the authors and do not necessarily reflect the views of the National Science Foundation. The funders had no role in study design, data collection and analysis, decision to publish, or preparation of the manuscript.

### Grant Disclosures

The following grant information was disclosed by the authors:
Advanced Research Projects Agency-Energy 'Electrofuels' Collaborative Agreement: DE-AR0000079.
National Science Foundation Graduate Research fellowship.
Ruth L. Kirschtein National research Service Award program of Harvard Catalyst.
The Harvard Clinical and Translational Science Center Award: UL1 RR 025758.
Harvard University.
National Center for Research Resources.
National Institutes of Health.

### Competing Interests

The authors declare there are no competing interests.

### Author Contributions

- Tyler J. Ford conceived and designed the experiments, performed the experiments, analyzed the data, contributed reagents/materials/analysis tools, wrote the paper, prepared figures and/or tables, reviewed drafts of the paper.
- Jeffrey C. Way analyzed the data, wrote the paper, reviewed drafts of the paper.

### Supplemental Information

Supplemental information for this article can be found online at http://dx.doi.org/10.7717/peerj.1040#supplemental-information.

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
