# Peer review of "Enhancement of E. coli acyl-CoA synthetase FadD activity on medium chain fatty acids"

_PeerJ, doi:10.7717/peerj.1040_

## Round 0.1 · original submission · Minor Revisions

The manuscript from Ford and Way reports on the generation and characterisation of FadD mutants to improve acyl-CoA synthetase activity on fatty acids. The effects of mutations are tested on partially purified variants as well as by following growth rates of clonal E. coli cultures.

As indicated by both reviewers, the experimental part of the paper is sufficiently solid and accurately described for the paper to be accepted for publication.

To improve clarity and significance further, I recommend that the authors incorporate a response to the questions raised by reviewer 2 (see 'Validity of the findings' section in the reviewer's comments) to explain the differences and complementarity between in vitro and in vivo assays.

The manuscript should also be revised for typos (as indicated by reviewer 1), while the presence of plasmids and primers tables could be reconsidered. Although these are in principle useful, listing internal reference names (not used anywhere else in the manuscript) and repetitive explanations of the nature of each primer/plasmid is redundant. If the authors prefer to keep such tables, I then encourage them to release the sequence of the FadD gene as well, indicate for each primer the mismatched bases, and move everything to the (already existing) SI.

Finally, I echo reviewer 2's comment regarding the possible applications of isolated mutants for biofuel production: the analysis reported in the present manuscript cannot be used to support such statements and far-fetched speculations should be kept to a minimum.

Considering the nature of the reviewers' comments, I expect to assess a revised version of the manuscript soon.

Reviewer 1 ·

Basic reporting

This report fulfils the basic standards of scientific reporting of work of this nature. Some corrections must be applied: e.g. 14,000 g on line 151 is probably rpm; imidazole is misspelt throughout this page also; The quality of Figures is acceptable; standard errors have been reported for relevant experiments such as that illustrated in Figure 1. Legends are full and informative; Literature has been cited appropriately, but check this section of errors in notation (capitilsation of journals etc.). Primer equences (of which there are many) could be relegated to a Supporting Information section (if relevant)

Experimental design

The experimental design follows a logical line of investigation; from observation of the natural behaviour through to mutation of relevant enzymes (some of this based on modelling) and their analysis.

Validity of the findings

The authors report enhanced activity of the fadD enzyme in reponse to mutations in the product exit tunnel. While one may quibble with the use of models, the results are suported by the experiemental evidence to a degree that permits the level of speculation provided in the discussion.

Additional comments

As above

Reviewer 2 ·

Basic reporting

While the study has merits the manuscript pitches a story about biofuel and biochemical production and the role of FadD in a potential future scenario. However the main phentotype measured is growth of E. coli. No attempt is made to show how this engineered enzyme can be used to make value added products. Thus it is not clear what significance these results have on the ability of E. coli expressing FadD mutants to be a better biocatalyst/producer of valuable products.
Therefore the manuscript is predominantly about the effect of mutations in FadD on the growth of E. coli and also to a lesser extent on the structure of fadD

Experimental design

For the most part the experiemntal design is strong.
An opportunity was missed to measure kinetics of SDM mutants where various amino acids were replaced with glycine. Measuring the effect on growth rate and not kinetics of the enzyme makes the study incomplete.

Validity of the findings

Some of the growth and enzyme kinetic data do not match (Figure 2 and 3) and there is a question mark around the validity of enzyme activity data for octanoate.

Figure 2 shows improvement of growth rate of E. coli with hexanoate and octanoate and a marginal positive effect with decanoate as a growth substrate. There is a corresponding improvement in activity of the enzyme with hex and octanoate (Figure 3). There are some contradictions in the growth and kinetic data e.g. mutant H376R has no statistically significant effect on growth rate with decanoate in Figure 2 but it has a statistically significant effect on activity of FadD with decanoate? Can this be explained?

Figure 3. The improvements in enzyme activity are statistically analyzed but the standard error for these assays is large and the authors state that the error for enzyme activity measurement with octanoate is unreliable. The effect on decanoate is marginal and the effect of activity towards heaxanoate was not shown (Figure 3).

Additional comments

Enhancement of E. coli acyl-CoA synthetase FadD activity on medium chain fatty acids.

The manuscript describes the in vitro mutagenesis of the FadD from E. coli. The use of directed evolution and site directed mutagenesis is described. The effect of these mutations on the growth rate of E. coli when the fadD genes were expressed on a plasmid was determined. Furthermore the effect of the mutations on enzyme activity and affinity was also determined. The replacement of amino acids at a channel where substrates/co-substrates and products/co-products entered and left the active site was undertaken to reduce side chain interference in this channel.

---

## Round 0.2 · accepted · Accept

I am satisfied with how the authors addressed the minor points made previously by the reviewers. I am happy with the current version and thus consider it acceptable for publication.

Reviewer 1 ·

Basic reporting

All clarifications and corrections requested by this reviewer have been addressed.

Experimental design

All clarifications and corrections requested by this reviewer have been addressed.

Validity of the findings

All clarifications and corrections requested by this reviewer have been addressed.